# Electrochemical Behavior of Some Cinchona Alkaloids Using Screen-Printed Electrodes

**DOI:** 10.3390/s25072216

**Published:** 2025-04-01

**Authors:** Tonino Caruso, Laura Palombi

**Affiliations:** 1Department of Chemistry and Biology, University of Salerno, 84084 Fisciano, Italy; 2Department of Physical and Chemical Sciences, University of L’Aquila, 67100 Coppito, Italy; laura.palombi@univaq.it

**Keywords:** voltammetric sensors, screen-printed electrodes, cinchona alkaloids, cathodic deposition, electrode surface coating

## Abstract

An effective deposition of a cinchonine layer on a platinum metal surface can be easily achieved through the cathodic reduction of a cinchonine hydrochloride methanolic solution at a controlled potential of −220 mV vs. the silver standard electrode (SSE). A coated screen-printed platinum electrode has proven to be suitable for cinchonine determination in water, urine, and serum at µg L^−1^ concentration levels using differential pulse voltammetry in a phosphate buffer solution (pH 7.0). The limits of detection (LOD) and quantitation (LOQ) were 0.6 µg L^−1^ and 1.8 µg L^−1^, respectively.

## 1. Introduction

In recent years, screen-printed electrodes (SPEs) have undergone remarkable advancements, with numerous scientific papers published on sensors and biosensors. SPEs offer several advantages over conventional electrodes, as they can be efficiently used for microvolumes and as single-use sensors, eliminating the need for electrode maintenance. This has paved the way for the development of reproducible, cost-effective, precise, and highly sensitive sensors. Thus, SPEs are ideal for decentralized assays and the development of (bio)sensors in environmental [1,2], clinical [3], and agri-food applications, as well as for electrosynthesis and compound characterization [4,5]. On the other hand, to enhance selectivity and/or sensitivity in various applications, appropriate compounds can be used to modify the electrode surface [6,7,8,9,10,11], including that of SPEs [12,13]. However, weak adsorption, particularly of organic molecules onto the electrode surface, has traditionally posed a challenge, limiting the effectiveness of electrochemical techniques for organic system analysis [14,15,16,17]. In this study, electrochemical deposition was employed to create a cinchonine-based organic thin film on a platinum electrode surface via the cathodic reduction of a cinchonine hydrochloride methanolic solution. Cinchonine (CN) is an alkaloid isolated from *Cinchona succirubra* and, when combined with other Cinchona alkaloids such as quinine, quinidine, and cinchonidine, plays a crucial role in pharmacological activity [18]. It is widely used as an important antimalarial agent [19,20,21], exhibits antiarrhythmic properties [22], provides resistance against various types of tumors [23], and acts as a potent inhibitor of human platelet aggregation [24,25]. Notably, cinchonine exhibits lower toxicity and higher activity compared to other quinine-related compounds [26]. Its lethal dose (LD_50_) in humans is 456 mg kg^−1^, with an estimated median lethal dose of 152 mg kg^−1^ [27]. Numerous analytical studies have focused on the determination of cinchonine-type alkaloids, both due to their medicinal and commercial significance and because of their narrow therapeutic window between ineffective and toxic doses. Standard quantification methods for the four main Cinchona alkaloids and their formulations rely exclusively on spectrophotometry [28,29], which lacks sensitivity, or liquid chromatography [30,31], which is time-consuming. To develop more efficient, sensitive, selective, and, above all, faster methods, various alternative approaches have been explored, including electroanalytical techniques [32,33,34,35,36,37,38]. In this study, we present a novel electrochemical sensor for determining cinchonine concentrations in the microgram-per-liter range, offering faster and more sensitive analysis compared to conventional techniques. The proposed sensor leverages the strong adsorption affinity of platinum for cinchonine while benefiting from the advantages of screen-printed electrodes. The adsorption of various cinchona alkaloids onto platinum surfaces has been extensively studied [39,40,41], with some exhibiting irreversible adsorption, requiring relatively high potentials for desorption [42]. As illustrated in Figure 1, the cathodic reduction of a methanolic cinchonine hydrochloride solution induces a hydrogen evolution reaction, leading to the stable adsorption of the insoluble alkaloid on the platinum cathode surface [43].

This modified electrode was employed to quantify the preferentially adsorbed species, specifically cinchonine and quinidine, using conventional differential pulse voltammetry. Notably, an increase in sensitivity will be demonstrated compared to an unmodified screen-printed electrode. The results obtained with this sensor are presented and discussed in detail in this paper.

## 2. Materials and Methods

### 2.1. Reagents

All reagents used in this study were of analytical grade, and all solutions were prepared using ultrapure solvents purchased from Merck. Cinchonine, cinchonine monohydrochloride hydrate, quinine hydrochloride dihydrate, and quinidine hydrochloride monohydrate were all obtained from Merck, while cinchonidine hydrochloride was purchased from Alfa Chemistry. *N*,*O*-Bis(trimethylsilyl)trifluoroacetamide (BSTFA) for GC derivatization and CCl_4_, suitable for IR spectroscopy were acquired from Supelco-Merck. All human serum and urine samples were obtained from the University Affiliated Hospital.

### 2.2. Instrumentation and Apparatus

Differential pulse voltammetry (DPV) and chronoamperometry (CA) were performed using an electrochemical system comprising an Autolab PGSTAT302F potentiostat/galvanostat (EcoChemie, Utrecht, The Netherlands) with an IME663 interface, connected to the 663 VA Stand system (Metrohm, Basel, Switzerland), and controlled by Nova 1.10.5 software (EcoChemie, Utrecht, The Netherlands). The stand was equipped with a screen-printed electrode holder (Metrohm, 6.1241.090) for connecting the working electrode, along with an external Ag/AgCl reference electrode (saturated with 3 mol L^−1^ KCl, SSE) and an external platinum auxiliary electrode, both from Metrohm. These components were used for cathodic deposition at a controlled potential. The working electrodes were commercially available screen-printed platinum (SP-Pt) electrodes (Aux.: Pt; Ref.: Ag), purchased from Metrohm-Dropsens (DRP-550 from Metrohm, Oviedo, Spain). Both potentiostatic and galvanostatic modes were employed for electrodeposition on the Pt surface of the SP-Pt working electrode. Additionally, an analytical balance with a sensitivity of 0.01 mg (Mod. XS105DR from Mettler-Toledo, Milan, Italy) was used for mass measurements, and a Julabo circulator (Mod. F12 from Thermo Fisher, Monza, Italy) was used to maintain a constant temperature. Fourier-transform infrared (FTIR) spectra were analyzed using an Agilent Cary 630 FTIR spectrophotometer (Agilent, Cernusco sul Naviglio, Italy) equipped with a ZnSe multi-bounce ATR sampling module. GC–MS analyses were performed using an Agilent 7890A GC (Agilent Technologies, Cernusco sul Naviglio, Italy) system coupled with an Agilent 5975 MSD (Agilent Technologies, Cernusco sul Naviglio, Italy) system.

### 2.3. Procedures

#### 2.3.1. Solutions

A 4.00 mg L^−1^ cinchonine standard solution was prepared by dissolving 4.50 ± 0.01 mg of cinchonine monohydrochloride hydrate (99%) in distilled water to a final volume of 10.0 mL, followed by vortexing for 5 min. The working standard solution was obtained through successive dilutions. As previously reported, a standard solution can also be prepared by dissolving cinchonine in an acidic solution [44]. Solutions of the other alkaloids were prepared in a similar manner. For DPV measurements, a solution containing 0.02 mol dm^−3^ phosphate buffer (pH 7.0) and 0.1 mol dm^−3^ KClO_4_ was used as a blank (electrolyte solution).

#### 2.3.2. Electrode Preparation

Electrochemical deposition was performed using a computer-controlled potentiostat (Autolab PGSTAT302F, from EcoChemie, Utrecht, The Netherlands) with a three-electrode cell, consisting of a Pt bar as the counter electrode and an Ag/AgCl electrode as the silver standard reference electrode (SSE). The working electrode was a screen-printed platinum electrode (SP-Pt). During deposition, only the WE contact of the screen-printed electrode holder was connected to the Autolab station. A white organic layer was deposited on the SP-Pt electrode surface through the cathodic reduction of alkaloid-derived ammonium salt solutions, performed at a controlled deposition potential of −0.220 V vs. SSE for 60 s [45]. The deposition solution consisted of 3 mg mL^−1^ cinchona alkaloid hydrochloride in methanol. The process was carried out at a constant temperature of 10 °C. After deposition, the screen-printed platinum electrode coated with alkaloid (SP-Pt/CN for cinchonine) was thoroughly rinsed with methanol. To confirm the successful deposition of cinchona, the deposited layer was removed by immersing a coated electrode in a vial containing approximately 2 mL of carbon tetrachloride [46], which was then placed in an ultrasonic bath for 10 s. This process was repeated with additional SP-Pt/CN electrodes to further concentrate the solution. The resulting organic solution was then deposited drop by drop onto the ZnSe crystal of the ATR module in the spectrophotometer. The recorded FTIR-ATR spectrum matched that of cinchonine (Appendix A). A similarly prepared solution was treated with BSTFA/Et_3_N and injected into the GC-MS to verify the purity and composition of the deposit. GC-MS analysis confirmed that the cathode deposit consists exclusively of cinchonine (Appendix A).

#### 2.3.3. Differential Pulse Voltammetry Measurements

For DPV measurements, 10.0 mL of the blank solution was placed in a glass voltammetric cell from Metrohm. The DPV experimental parameters were set as follows: scan range from +0.20 V to +1.10 V, pulse amplitude of 50 mV, pulse time of 50 ms, and scan rate of 10 mV s^−1^. For methodology validation, cinchonine quantification was performed in different samples. The standard addition method was used for calibration, with calibration plots obtained through successive additions of cinchonine standard solution into the blank solution. All experiments were conducted at a controlled temperature of 25.0 °C using a thermostatic cell. Prior to electrochemical measurements, solutions were degassed with pure nitrogen.

#### 2.3.4. Data Treatment

All data analyses were conducted and interpreted using NOVA 1.10.5 software, while additional calculations were performed with the EXCEL^®^ program. Sensitivities were expressed as the slope of the calibration plot. Limits of detection (LODs) and limits of quantification (LOQs) were calculated as three and ten times the standard deviation of the intercept of the calibration line, respectively, divided by the slope [47,48]. Repeatability was evaluated through ten consecutive measurements at a concentration within the middle of the linear range, while reproducibility was determined from the relative standard deviation (RSD) of the slopes of three independent calibration plots.

## 3. Results and Discussion

### 3.1. Preliminary Studies

It is well known that cinchona alkaloids adsorb onto the Pt surface [40] and that the two asymmetric carbon atoms influence the chirality of hydrogenation products [49]. Preliminary experiments were conducted on platinum electrodes at different fixed current densities, ranging from 1.0 mA to 10 μA. These experiments enabled the deposition of alkaloid layers on the Pt electrode surface; however, the deposits were later found to be poorly adherent, likely due to the simultaneous and uncontrollable evolution of hydrogen. Similar deposits were observed even when deposition was carried out at low temperatures. Based on these preliminary results, further experiments were performed using an SP-Pt electrode at a controlled potential to mitigate hydrogen evolution [50,51] during deposition on the Pt electrode surface. Various deposition potentials were investigated, ranging from 0 to −500 mV vs. SSE. Figure 1a illustrates the relationship between the applied potentials and the current densities (μA cm^−2^) measured during cinchonine electrodeposition at controlled potential. Deposition times of 30, 60, 90, and 120 s were tested. As shown in Figure 1a, significant current consumption due to H_2_ evolution occurs when a more cathodic deposition potential is applied (E_app_ < −300 mV) [52]. This suggests that cinchonine chemisorption can be effectively achieved under controlled potential conditions at moderately negative potentials, where hydrogen evolution is negligible. To further analyze this, we plotted the deposition potential against log current density (Figure 1b). The graph shows a linear trend above −220 mV, leading to the selection of this deposition potential for subsequent experiments [53]. Appendix A shows the chronoamperometric curves recorded at different deposition potentials.

To improve the homogeneity of the deposited layer, depositions were carried out at low temperatures, ranging from 20 °C to 5 °C [54]. To this end, a series of SP-Pt/CN electrodes was prepared under potential control at different temperatures (20 °C, 15 °C, 10 °C, and 5 °C) and evaluated for reproducibility and stability as probes for differential pulse voltammetric determinations. The stability of the modified SP-Pt/CN electrodes obtained at different temperatures was examined through multiscan studies to identify the optimal deposition conditions. The best electrode was determined based on the smallest relative standard deviation (RSD) in peak current [55]. In these studies, ten consecutive scans were performed in a 20.0 μg L^−1^ cinchonine solution. This test also helped assess whether the chemisorbed species would diffuse from the electrode surface into the bulk solution during continuous measurements under the same conditions. No significant decrease in peak currents, which could indicate the diffusion of some particles from the electrode interface into the solution, was observed during the multiscan test. Instead, only a variation in peak current values was detected, with a calculated RSD ranging from 1.1% to 6.4% (Table 1). These results confirm that the electroactive material remains confined to the electrode surface and does not detach into the bulk solution. Since the RSD value for electrochemical methods and stationary printed electrodes should be below 5%, the reproducibility of the developed method is considered acceptable under most of our experimental conditions [35].

In particular, the deposition conditions achieved at 10 °C for 60 s, at a controlled potential of −220 mV vs. SSE, resulted in the lowest RSD value (1.1%), as shown in Figure 2. Therefore, the developed sensor demonstrates good stability and reproducibility for cinchonine detection and will be used to quantify cinchonine in real samples, such as urine and plasma. The advantage of this modified SPE/Pt is that it can be regenerated after approximately ten uses. Simply immersing it in methanol in an ultrasonic bath for 10 s and repeating the cathodic deposition process is sufficient for regeneration.

### 3.2. Electrode Testing

Pulse voltammetric techniques, such as DPV, are effective and rapid electroanalytical methods with well-established advantages, including excellent discrimination against background current and low detection limits [56]. To demonstrate the sensitivity of SP-Pt/CN electrodes for the electrochemical measurement of cinchona alkaloids, the effect of varying cinchonine concentration using DPV mode was investigated. The standard addition method was employed for calibration. Differential pulse voltammograms and the corresponding calibration plots, shown in Figure 3, were obtained through successive additions of a cinchonine standard solution (4.00 mg L^−1^) into a blank solution (10.0 mL). Considering the physiological conditions of the human body, a pH of 7.0 was selected, as this value prevents cinchonine protonation and ensures a stable sensor response with cinchonine adsorbed on its surface [33]. As illustrated in Figure 3a, the screen-printed platinum electrode produced a weaker and less intense signal compared to the SP-Pt/CN electrode. At cinchonine concentrations below 4 μg L^−1^, the DPV signal is nearly indistinguishable from the blank voltammogram, suggesting slow electrode kinetics for the faradaic processes generated by cinchonine oxidation, despite the well-documented strong affinity of platinum for cinchonine [43]. In contrast, the SP-Pt/CN electrode exhibited a well-defined and more intense characteristic oxidation peak, even at concentrations as low as 1.3 μg L^−1^ (Figure 3b). This clearly indicates that the cinchonine layer enhances the sensitivity of the working electrode by promoting electrochemical processes on a larger and more specific surface area due to the adsorbed layer [16]. From the electrochemical current–potential responses shown in Figure 3a,b, the peak current (I_p_) was measured at various cinchonine concentrations. The resulting calibration plot showed a linear relationship with cinchonine concentration over a range of 1.3 μg L^−1^ to 39.6 μg L^−1^, following the regression equation: I_p_ (μA) = 4.228 c (μg L^−1^) + 4.812 with a correlation coefficient of 0.9996.

The limits of detection (LOD) and quantitation (LOQ) were calculated from the oxidation peak currents using the following equations [47]:LOD = 3 σ/mLOQ = 10 σ/m
where σ represents the standard deviation of the oxidation peak current (three runs), and m is the slope of the corresponding calibration curves. For the SP-Pt/CN electrode, the LOD and LOQ were found to be 0.6 μg L^−1^ and 1.8 μg L^−1^, respectively. In contrast, the calibration plots for the untreated SP-Pt electrode exhibited a linear relationship with cinchonine concentration only at concentrations above 12.0 μg L^−1^, with a regression equation: I_p_ (μA) = 0.716 c (μg L^−1^) + 1.435, and a correlation coefficient of 0.9987. Additionally, the untreated SP-Pt electrode had a lower slope than the SP-Pt/CN electrode. Since slope is a measure of sensitivity, a steeper line with a larger slope indicates a more sensitive measurement [57]. Indeed, for the untreated SP-Pt electrode, the LOD and LOQ decreased to 5.2 μg L^−1^ and 17.5 μg L^−1^, respectively. All data are summarized in Table 2, which compares the detection limits for cinchonine determination obtained using various sensors, including our modified electrode and conventional methods [29,32]. This study demonstrated a significantly lower detection limit compared to the untreated screen-printed Pt electrode, as well as to other modified sensors reported in the literature.

In this study, we also tested other cinchona alkaloids, including cinchonidine, quinine, and quinidine, considering that cinchonine and cinchonidine, as well as quinine and quinidine, are stereoisomers with the same molecular formulas and similar structures (Figure 4).

We found that only SP-Pt electrodes with electrodeposited cinchonine were sufficiently stable for subsequent determinations. The balance between the adsorption strength of these molecules on the Pt surface and their varying solubilities can explain the behavior of the different cinchona alkaloids. This observation is consistent with findings reported by Zaera, who clarified certain apparent contradictions in the literature regarding the effectiveness of these four cinchona alkaloids as chiral catalysis promoters [43]. No favorable electrochemical conditions were identified for the deposition of quinine and cinchonidine, likely due to their significantly higher solubility in methanol compared to the other two alkaloids [46]. Additionally, the electrode with deposited quinidine was found to be insufficiently stable for measurements in neutral aqueous solutions due to its greater solubility in this medium.

### 3.3. Selectivity

Several expected organic species were selected to evaluate the selectivity of the cinchonine sensor (Table 3). The effect of potential interferents on the electrochemical determination of cinchonine was investigated by fixing the initial cinchonine concentration at 6.0 μg L^−1^ (10× LOD) and the interferent concentrations at 0.6 μg L^−1^ (1:0.1), 6.0 μg L^−1^ (1:1), or 60 μg L^−1^ (10×). Subsequently, varying amounts of cinchonine were added to the initial solution, and recovery studies conducted using the standard addition method showed satisfactory recovery rates ranging from 91.8% to 109.6% (Table 3). This indicates that all tested species caused no interference when present in concentrations up to 10 times higher than that of cinchonine, with the exception of quinidine. This behavior can be attributed to the structural similarity between quinidine and cinchonine, differing only by the presence of an additional methoxy group at the C6′ position of the quinoline ring (Figure 4). However, quinidine does not interfere when its concentration is less than 1/10 that of cinchonine. Additionally, the SP-Pt/CN electrode was successfully used for the determination of quinidine in the absence of cinchonine.

### 3.4. Application of the Sensor: Determination of Cinchonine in Urine and Serum

The proposed SP-Pt/CN electrode was successfully applied for the determination of cinchonine in serum and urine samples obtained from the University Affiliated Hospital. The cinchonine concentration in human serum reaches its peak within 1–2 h after assimilation as a drug, which occurs rapidly and almost completely in the stomach and intestine. Approximately 3.6–6% of the total administered dose is excreted unchanged in the urine [58]. A series of sample solutions were prepared by adding an appropriate amount of cinchonine standard solution to 10.0 mL of serum or urine. Then, 1000 μL of each solution was transferred into the voltammetric cell and diluted to 10.0 mL with buffer solution. Subsequently, specific volumes of cinchonine standard solution were added using a micro-syringe, and differential pulse voltammograms (DPVs) were recorded after each standard addition. The results, summarized in Table 4, demonstrate that the SP-Pt/CN electrode enables the reliable determination of cinchonine, yielding satisfactory recovery values. Freshly prepared SP-Pt/CN electrodes can be stored before testing, as they remain stable.

## 4. Conclusions

In this study, we demonstrate that cinchonine can be effectively deposited on a screen-printed platinum surface via the cathodic reduction of its hydrochloride salt dissolved in a methanolic solution. A stable cinchonine layer is formed when the deposition is performed for 60 s at a controlled potential of −220 mV vs. the silver standard electrode, with the temperature maintained at 10 °C. The repeatability and reproducibility of the modified electrode were assessed by calculating the relative standard deviation (RSD) of the peak current values from 10 consecutive DPV measurements (multiscan studies) conducted at a cinchonine concentration of 20.0 μg L^−1^, which lies in the middle of the linear range. The obtained RSD value confirms good reproducibility and indicates that the electrode surface remains stable during voltammetric measurements with the coated screen-printed electrode. Modified screen-printed platinum electrodes were successfully employed for cinchonine determination at physiological pH 7.0. The proposed sensor, which leverages the strong adsorption tendency of platinum for cinchonine along with the advantages of screen-printed electrodes, exhibited a very low detection limit and a wide linear dynamic range. The limits of detection (LOD) and quantification (LOQ) were 0.6 μg L^−1^ and 1.8 μg L^−1^, respectively, which are significantly lower than those reported for other conventional methods. Finally, the applicability of our sensor for clinical assays of cinchonine in human serum and urine was investigated, yielding promising results.

## Data Availability

Data are contained within the article and Appendix A.

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
