# Peer review of "Electrochemical Behavior of Some Cinchona Alkaloids Using Screen-Printed Electrodes"

_sensors, 2025, doi:10.3390/s25072216_

Round 1
Reviewer 1 Report
Comments and Suggestions for Authors
This article presents a study on the electrochemical detection of cinchona alkaloids using SPE. In general, I think this article is scientifically sound and could be considered to be accepted for publication. However, there are some issues that need to be addressed by the authors before it can be accepted. Therefore, I recommend the article can only be considered after major revision. My comments are presented below:
- The current English proficiency should be enhanced. There are several grammatical errors and unclear sentences found in the manuscript, which could lead to misinterpretation. I strongly recommend the author(s) to submit their manuscript to native English speaker for better readership of the article.
- In the introduction section, it is still unclear why detection of cinchona alkaloid is essential. I strongly suggest the authors to elaborate more on the significance for cinchona detection, what is the current conventional method of detection, what is the current drawback of such methods, and what is this study offer to solve such drawbacks.
- Scheme 1 is somewhat confusing and potentially misleading to the readers. Typically, SPE and electrochemical detection in general are carried out in three electrode system. Please revise.
- Surface chemistry of the electrode is typically greatly influence the electrochemical behavior of the electrode. Therefore, I suggest the authors to provide FESEM images and EDS mapping of the electrode before and after the deposition.
- Figure 1 and 2 are best combined into one figure. Additionally, during the deposition, I think it is essential to showcase the plot for chronoamperogram to provide comparison on the current density of each experiment.
- FTIR or Raman spectroscopy analysis are also essential to confirm that deposition of cinchona was really happening.
- For detection parameters, it is also essential for the authors to compare their result (selectivity, LOD, and LOQ) with both gold standard of cinchona quantification (using HPLC or other methods) and those reported elsewhere.

Comments on the Quality of English Language
The current English proficiency should be enhanced. There are several grammatical errors and unclear sentences found in the manuscript, which could lead to misinterpretation. I strongly recommend the author(s) to submit their manuscript to native English speaker for better readership of the article.
Author Response
Comments 1: The current English proficiency should be enhanced. There are several grammatical errors and unclear sentences found in the manuscript, which could lead to misinterpretation. I strongly recommend the author(s) to submit their manuscript to native English speaker for better readership of the article.
Response: 1: Thank you for pointing that out. I sincerely apologize. I have sent the manuscript to a native English speaker for any necessary refinements.
Comment 2: In the introduction section, it is still unclear why detection of cinchona alkaloid is essential. I strongly suggest the authors to elaborate more on the significance for cinchona detection, what is the current conventional method of detection, what is the current drawback of such methods, and what is this study offer to solve such drawbacks.
Response 2: Thank you very much for this suggestion. I have revised the introduction to highlight the importance of Cinchona alkaloids and to enhance the comparison with conventional techniques, such as chromatographic methods, which, however, require longer analysis times, and spectroscopic methods, which, however, exhibit lower sensitivity. Table 2 and the references have also been updated accordingly. (page 2, paragraph 1; pages 7 and 8, paragraph 3.2).
Comment 3: Scheme 1 is somewhat confusing and potentially misleading to the readers. Typically, SPE and electrochemical detection in general are carried out in three electrode system. Please revise.
Response 3: Thank you for pointing this out. I have revised Scheme 1 to focus exclusively on the cathodic deposition process, which is the key aspect of the method used for electrode preparation.
Comment 4: Surface chemistry of the electrode is typically greatly influence the electrochemical behavior of the electrode. Therefore, I suggest the authors to provide FESEM images and EDS mapping of the electrode before and after the deposition.
Response 4: I apologize, but my department lacks the necessary instrumentation to carry out these experimental determinations. However, the electrochemical behavior has been demonstrated nonetheless.
Comment 5: Figure 1 and 2 are best combined into one figure. Additionally, during the deposition, I think it is essential to showcase the plot for chronoamperogram to provide comparison on the current density of each experiment.
Response 5: As per your kind advice, Figures 1 and 2 have been combined into a single figure. The chronoamperogram plots have been included in the Supplementary Materials..
Comment 6: FTIR or Raman spectroscopy analysis are also essential to confirm that deposition of cinchona was really happening
Response 6: Thank you for your advice. Following the recommendations in the literature (Zaera), I removed the deposited layer and recorded an ATR spectrum to confirm its nature. This information has been incorporated into the text (page 3, paragraphs 2.2 and 2.3.2). The FTIR-ATR spectra of pure and deposited cinchonine have been inserted in the Supplementary Materials.
Comment 7: For detection parameters, it is also essential for the authors to compare their result (selectivity, LOD, and LOQ) with both gold standard of cinchona quantification (using HPLC or other methods) and those reported elsewhere.
Response 7: Thanks for pointing this out. I have enhanced the information in Table 2 by expanding the comparison, which was previously limited to sensors, to also include conventional spectrophotometric and chromatographic techniques (pages 7-8, paragraph 3.2).
Reviewer 2 Report
Comments and Suggestions for Authors
In this manuscript, the authors developed cinchonine-deposited screen-printed Pt electrodes for cinchonine detection, and demonstrated good sensitivity and selectivity both in standard and clinically relevant solutions. I think this work is well-executed and with good analysis. With the minor concern addressed, I believe it is suitable for publication.
- On line 169. Could the authors clarify the peak current change for the SP-Pt/CN stability test? Since there was no current decrease, did the peak current increase for the 10 scans? Additionally, including a legend in Figure 3, such as an arrow and a label indicating “Scan 1 to 10,” would enhance clarity.
- Sections “Selectivity” and “Application of Sensor:…” are both numbered as 3.3.
- Could the authors provide further details on the test conditions involving urine and serum? Were the urine and serum samples clinical, or simulated? Were the SP-Pt/CN electrodes stored before testing, or were they freshly prepared? Since these conditions are of great interest for clinical applications.
Author Response
Comment 1: On line 169. Could the authors clarify the peak current change for the SP-Pt/CN stability test? Since there was no current decrease, did the peak current increase for the 10 scans? Additionally, including a legend in Figure 3, such as an arrow and a label indicating “Scan 1 to 10,” would enhance clarity.
Response 1: The peak current change observed during the stability test corresponds to 10 successive scans of the same sensor at a fixed cinchonine concentration. These values fluctuate with an RSD of 1.1%, indicating good performance of the SP-Pt/CN electrode. No monotonic behavior is expected.
Comment 2: Sections “Selectivity” and “Application of Sensor:…” are both numbered as 3.3.
Response 2: Thank you. I apologize for the mistake. It has been corrected.
Comment 3: Could the authors provide further details on the test conditions involving urine and serum? Were the urine and serum samples clinical, or simulated? Were the SP-Pt/CN electrodes stored before testing, or were they freshly prepared? Since these conditions are of great interest for clinical applications.
Response 3: Thank you for pointing out these details. As stated (page 2, section 2.1), all human serum and urine samples were obtained from the University Affiliated Hospital. The SP-Pt/CN electrodes can be stored before testing, as they are stable, or they can be freshly prepared. I have clarified these points in the text.
Reviewer 3 Report
Comments and Suggestions for Authors
The manuscript presents a well-structured study on cinchonine determination by DPV. However, there are several areas that require improvement. Below, I outline some concerns that should be addressed before publication.
- The cinchonine should preferably be labelled next to the molecular formula in Figure 1.
- The author claimed that “An appropriate assay analysis method is needed to determine an efficient, effective, and selective drug dose”. They should explain how the proposed sensor can be applied in drug dosage studies.
- Straight lines should be shown in Figure 2.
- The basic mechanism is cathodic deposition. The working electrode is disposable or be used for several times, does the deposition affect the subsquent detection?
Author Response
Comment 1: The cinchonine should preferably be labelled next to the molecular formula in Figure 1.
Response 1: I agree. Therefore, I have modified Figure 1, also taking into account the suggestions of the other reviewers.
Comment 2: The author claimed that “An appropriate assay analysis method is needed to determine an efficient, effective, and selective drug dose”. They should explain how the proposed sensor can be applied in drug dosage studies.
Response 2: I agree. Accordingly, I have modified the text to emphasize this point in the discussion of this sensor’s applications for the quantification of cinchonine in serum and urine (page 9, paragraph 3.4).
Comment 3: Straight lines should be shown in Figure 2.
Response 3: Thank you for pointing this out. I have modified Figure 2 by adding straight lines.
Comment 4: The basic mechanism is cathodic deposition. The working electrode is disposable or be used for several times, does the deposition affect the subsquent detection?
Response 4: The developed sensor demonstrated good stability and reproducibility for cinchonine detection. The modified screen-printed electrode can be used up to 10 times, as confirmed by stability studies. After that, it needs to be regenerated. I have added this clarification to the text. (page 6, paragraph 3.1).
Round 2
Reviewer 1 Report
Comments and Suggestions for Authors
The authors have revised the manuscript according to the suggestions and comment. I think it now can be accepted.